# Automated tail-informed threshold selection for extreme coastal sea levels

Thomas P. Collings<sup>1</sup>, Callum J. R. Murphy-Barltrop<sup>2,3</sup>, Conor Murphy<sup>4</sup>, Ivan D. Haigh<sup>1,5</sup>, Paul D. Bates<sup>1,6</sup>, and Niall D. Quinn<sup>1</sup>

**Correspondence:** Thomas P. Collings (t.collings@fathom.global)

**Abstract.** Peaks over threshold (POT) techniques are commonly used in practice to model tail behaviour of univariate variables. The resulting models can be used to aid in risk assessments, providing estimates of relevant quantities such as return levels and periods. An important consideration during such modelling procedures involves the choice of threshold; this selection represents a bias-variance trade-off and is fundamental for ensuring reliable model fits. Despite the crucial nature of this problem, most applications of the POT framework select the threshold in an arbitrary manner and do not consider the sensitivity of the model to this choice. Recent works have called for a more robust approach for selecting thresholds, and a small number of automated methods have been proposed. However, these methods come with limitations, and currently, there does not appear to be a 'one size fits all' technique for threshold selection. In this work, we introduce a novel threshold selection approach that addresses some of the limitations of existing techniques, which we have termed the *Tail-informed threshold selection* (TAILS) method. In particular, our approach ensures that the fitted model captures the tail behaviour at the most extreme observations, at the cost of some additional uncertainty. We apply our method to a global data set of coastal observations, where we illustrate the robustness of our approach and compare it to an existing threshold selection technique and an arbitrary threshold choice. Our novel approach is shown to select thresholds that are greater than the existing technique. We assess the resulting model fits using a right-sided Anderson-Darling test, and find that our method outperforms the existing and arbitrary methods on average. We present and discuss, in the context of uncertainty, the results from two tide gauge records; Apalachicola, US, and Fishguard, UK. In conclusion, the novel method proposed in this study improves the estimation of the tail behaviour of observed coastal water levels, and we encourage researchers from other disciplines to experiment using this method with their own data sets.

## 1 Introduction

Natural hazards such as flooding, earthquakes and wildfires devastate communities and livelihoods around the world. Extreme value analysis (EVA) applied to the historical records of such events provides a useful tool for describing the frequency and

<sup>&</sup>lt;sup>1</sup>Fathom, Floor 2, Clifton Heights, Clifton, Bristol BS8 1EJ, UK

<sup>&</sup>lt;sup>2</sup>Technische Universität Dresden, Institut Für Mathematische Stochastik, Dresden, Germany

<sup>&</sup>lt;sup>3</sup>Center for Scalable Data Analytics and Artificial Intelligence (ScaDS.AI), Dresden/Leipzig, Germany

<sup>&</sup>lt;sup>4</sup>School of Mathematical Sciences, Lancaster University, Lancaster, LA1 4YF, UK

<sup>&</sup>lt;sup>5</sup>School of Ocean and Earth Science, University of Southampton, National Oceanography Centre, European Way, Southampton SO14 3ZH, UK

<sup>&</sup>lt;sup>6</sup>School of Geographical Sciences, University of Bristol, Bristol BS8 1SS, UK

intensity of these processes, and can be used by practitioners, community leaders, and engineers to prepare in advance for catastrophic events. Example applications include flood risk assessment (D'Arcy et al., 2023), nuclear regulation (Murphy-Barltrop and Wadsworth, 2024), ocean engineering (Jonathan et al., 2014), and structural design analysis (Coles and Tawn, 1994). Furthermore, stakeholders with assets spread across large geographical regions also utilise these tools to understand the hazard hazards across regional, continental, and global scales; see, for instance, Keef et al. (2013), Quinn et al. (2019), and Wing et al. (2020) (Keef et al., 2013; Quinn et al., 2019; Wing et al., 2020).

Coastal flood events, driven by high tides, surges, or waves, are commonly recorded at tide gauge stations, which cover large proportions of the populated global coastline. When characterising extreme sea level events, these tide gauge records are a primary source of information available to coastal managers. Due to the large number of sites involved, automated techniques for the characterisation of extreme events are preferable.

The earliest EVA techniques used the annual maximum approach, whereby a theoretically motivated distribution is fitted to the observed yearly maxima. However, this approach suffers from the drawback that only one observation is recorded for each year, resulting in some extreme observations being disregardedinefficient use of the data. In practice, this can lead to an incomplete picture of the upper tail , and consequently, and less accurate estimates of tail quantities, such as return levels. Consequently, recent consensus has been to move away from the annual maximum approach (Pan and Rahman, 2022) (Davison and Smith, 1990; Coles, 2001; Scarrott and MacDonald, 2012; Pan and Rahman, 2022).

As a result, the POT peaks over threshold (POT) approach has become the most popular technique for EVA modelling; see Section 3 and Coles (2001) for further details. This approach involves fitting a statistical model to data above some high threshold. However, the choice of this threshold is not arbitrary consequence-free, and inappropriate choices can result in poor model fits and extrapolation into the tail. Traditional approaches rely on visual assessments of parameter stability above the appropriate threshold. Such approaches suffer from subjectivity (Caballero-Megido et al., 2018) and the time input required to apply such techniques to global tide gauge records is not feasible. Consequently, many efforts have been made to reduce the time burden incurred by manual threshold selection. These include simplifications that allow large amounts of data to be processed, but at the cost of accuracy, e.g., using a static threshold, such as the 0.98 quantile or a fixed number of exceedances per year (Hiles et al., 2019; Collings et al., 2024). We refer to the approach of selecting a static 0.98 quantile across all sites (or variables) as the Q98 approach henceforth. Other approaches aim to automate much of the subjective decision-making process while retaining a flexible method that can capture the underlying behaviour of the physical processes (Solari et al., 2017; Curceac et al., 2020; Murphy et al., 2025). However, as discussed in Section 3, many of these techniques are not sufficiently robust or flexible and can perform poorly in practice.

In this study, our aim is to build upon existing techniques to provide a novel approach to automating threshold selection, which is applicable to a wide range of datasets whereby the extremes are characterised by different drivers. As a motivating example, we apply our method to a global dataset of 417 tide gauge records, demonstrating the performance of our approach over a variety of locations and benchmarking against other commonly used techniques.

50

The layout of this paper is as follows; in Section 2 we introduce the dataset used in this study and in Section 3 we discuss the common difficulties in using the POT approach across such a large, varied dataset, as well as some of the methods used to

simplify the process. In Section 4, we describe our novel approach to automating threshold selection and explain the subjective choices we have made in the methodchoices of tuning parameters. In Section 5, we present the results of applying our method to the global tide gauge dataset described in Section 2. In Section 6, we discuss our results in the context of uncertainty, bias, and the underlying physical processes and finally. Finally, in Section 7, we provide a conclusion to our study.

#### 60 2 Data

The locations of the considered tide gauge stations are illustrated in Figure 1. These data are obtained from the Global Extreme Sea Level Analysis (GESLA) database (Haigh et al., 2023), version 3.1, which is a minor update to version 3 to include the most recent years (2022-2024). The GESLA database was collated from many organisations that collect and publish tide gauge data. The water level records are prepared using the quality control flags published by the authors Haigh et al. (2023) alongside the data set, and duplicate timestamps in the records are also removed. The water level records that contain over 40 years of good data (defined as at least 75% complete) are retained. This results in a total of 417 water level records from around the world, which have an average record length of 66 years. The raw time series data are provided on a range of time steps (10, 15, and 60 minutes), and so are interpolated resampled to hourly resolution. A linear trend is calculated and removed to account for mean sea level risechange. Daily maxima data are obtained from the hourly records, and the data is subsequently declustered using a 4-day storm window to ensure event independence (Haigh et al., 2016; Sweet et al., 2020). Given the range of oceans and coastlines covered, one would generally expect to observe a wide variety of tail behaviours across the records.

**Figure 1.** Map of GESLA record locations with record lengths greater than 40 years. The two locations highlighted in red are Apalachicola, US and Fishguard, UK, which are discussed in more detail in Section 5.4.

#### 3 POT modelling

The POT approach, whereby a theoretically motivated distribution is fitted to the excesses of some high threshold (see, e.g., Coles, 2001), is the most common technique for assessing tail behaviour in environmental settings. Given any random variable

X and a threshold u, the results of Balkema and de Haan (1974) and Pickands (1975) demonstrate that under weak conditions, the excess variable  $Y := (X - u \mid X > u)$  can be approximated by a *generalised Pareto distribution* (GPD) – so long as the threshold u is 'sufficiently large'. The GPD has the form

$$H(y;\sigma,\xi) = 1 - \left(1 + \frac{\xi y}{\sigma}\right)_{+}^{-1/\xi}, \quad y > 0,$$
(1)

where  $z_+ = \max(0, z)$ ,  $\sigma > 0$ , and  $\xi$  denotes any real number  $\xi \in \mathbb{R}$ . We refer to  $\sigma$  and  $\xi$  as the scale and shape parameters, respectively, and we remark that the latter parameter quantifies important information about the form of tail phenomena; see Davison and Smith (1990) for further discussion. A wide range of statistical techniques have been proposed, including both Bayesian and frequentist frameworks, to fit the model in equation (1) (Dupuis, 1999; Behrens et al., 2004; Searrott and MacDonald, 2012; Northrop (Dupuis, 1999; Behrens et al., 2004; Northrop et al., 2017), although we note that maximum likelihood estimation (MLE) remains the most common technique (e.g., Gomes and Guillou, 2015). Consequently, we restrict attention to MLE techniques throughout this paper.

85

In many practical contexts, equation (1) is used to obtain estimates of return levels for some return period N of interest. Such values offer a straightforward interpretation: the N-year year return level is the value  $x_N$  that one would expect to exceed once, on average, every N years. Return levels are easily obtained by inverting equation (1) (see Coles, 2001), and their estimates are often used to inform decision making. For example, in the contexts of flood risk analysis and nuclear infrastructure design, regulators typically specify design levels corresponding to return periods of N = 100 years (D'Arcy et al., 2023) and N = 10,000 years (Murphy-Barltrop, 2023), respectively.

The ambiguity of the statement 'a sufficiently large threshold u' requires careful consideration. This is a problem that is commonly overlooked in many applications, and selecting a threshold u is entirely non-trivial. In particular, this selection represents a bias-variance trade-off: selecting a threshold too low will induce bias by including observations that do not represent tail behaviour, while extremely high thresholds will result in more variability due to lower sample sizes. Furthermore, the estimates of return levels are very sensitive to the choice of threshold, and biased estimates can significantly impact the cost and effectiveness of certain infrastructures, such as flood defences (Zhao et al., 2024).

Owing to the importance of threshold choice, a plethora of methods have been proposed which aim to balance the aforementioned trade-off; see Belzile et al. (2023) for an extensive? for a recent review of the literature. The standard and most-widely used approach for threshold selection involves a visual assessment of the stability of the GPD shape parameter across a range of increasing thresholds (Coles, 2001). This approach suffers from subjectivity in the choice of stable region. Furthermore, visual assessments assessment for individual sites is simply not feasible (within a reasonable time scale) for large scale applications.

Automatic approaches seek to remove this subjectivity by selecting a threshold based on some criterion or goodness-of-fit metric; Wadsworth and Tawn (2012) and Northrop and Coleman (2014) utilise penultimate models and hypothesis testing; Bader et al. (2018) and Danielsson et al. (2019) use goodness-of-fit diagnostics; Wadsworth (2016) utilise a sequential assessment of a changepoint model; and Northrop et al. (2017) create a measure of predictive performance in a Bayesian framework. Tancredi et al. (2006) avoid the prior selection of the threshold by employing a Bayesian mixture model where the threshold is

estimated as part of the parameter estimation, allowing for straight-forward estimation of threshold uncertainty. In the applied literature, Durocher et al. (2018) and Curceac et al. (2020) compare several automated goodness-of-fit approaches for selecting an appropriate a threshold in the hydrological setting. Furthermore, Choulakian and Stephens (2001), Li et al. (2005) and Solari et al. (2017) automate goodness-of-fit procedures and apply these techniques to a range of precipitation and river flow data sets.

Recently, Murphy et al. (2025) proposed a novel threshold selection technique building on the work of Varty et al. (2021). This method, termed the *expected quantile discrepancy* (EQD), aims to select a threshold u for which the sample excesses are most consistent with a GPD model. We briefly outline this method below. Let  $\mathbf{x}_u = (x_1, \dots, x_{n_u})$  be the sample of excesses of some candidate threshold u, i.e., a sample from Y. For each candidate threshold, the EQD method assesses the expected deviation between sample and theoretical quantiles at a set of fixed probabilities  $\mathcal{P}_m := \{j/(m+1): j=1,\dots,m\}$ , where m denotes some large whole number. This assessment is done across a large number of bootstrapped samples, say B, to incorporate sampling variability and stablise the threshold choice. More specifically, letting  $\mathbf{x}_u^b$  denote the  $b^{th}$  bootstrapped sample of  $\mathbf{x}_u$ , with  $b=1,\dots,B$ , Murphy et al. (2025) propose the metric

$$d_b(u) := \frac{1}{m} \sum_{j=1}^m \left| \frac{\hat{\sigma}_u^b}{\hat{\xi}_u^b} \left[ \left( 1 - \frac{j}{m+1} \right)^{-\hat{\xi}_u^b} - 1 \right] - Q\left( \frac{j}{m+1}; \boldsymbol{x}_u^b \right) \right|, \tag{2}$$

where  $(\hat{\sigma}_u^b, \hat{\xi}_u^b)$  denote the GPD parameter estimates for  $x_u^b$ , obtained using MLE, and  $Q(j/(m+1); x_u^b)$  denotes the j/(m+1) empirical quantile of  $x_u^b$ . Considering equation (2) over each bootstrapped sample, an overall measure of fit for u is given by  $d(u) = \sum_{b=1}^B d_b(u)/B$ . Finally, the selected threshold,  $u^*$ , is the value that minimises d, i.e.,  $u^* := \arg\min d(u)$ . Through an extensive simulation study, alongside several case studies, Murphy et al. (2025) show that their approach convincingly outperforms the core existing approaches for threshold selection. They further find that existing techniques do not provide sufficient flexibility or robustness to select appropriate threshold choices across a wide range of datasets. Therefore, at the time of writing, the EQD technique is the best available approach for automating leading approach for automated threshold selection.

In this article, we argue and demonstrate that while the EQD approach appears to work well in a wide variety of cases, it can suffer from drawbacks in certain contexts that result in less than ideal threshold choices. Specifically, the chosen thresholds can result in model fits that do not match up well at the most extreme observations. We briefly explore the reasons for why this may occur below.

To begin, consider two candidate thresholds  $u_1 < u_2$  satisfying  $\Pr(X > u_1) = 0.5$  (i.e., the median) and  $\Pr(X > u_2) = 0.99$ . Taking each threshold in turn, the EQD computes quantiles from the (bootstrapped) conditional variables  $(X - u_1 \mid X > u_1)$  and  $(X - u_2 \mid X > u_2)$  that correspond with the probability set  $\mathcal{P}_m$ . When considered on the scale of the data, however, this results in very different quantile probabilities. Letting  $x_{u_1,j}$  denote the (true) j/(m+1) quantile of  $(X - u_1 \mid X > u_1)$  for

any  $j = 1, \dots, m$ , we have

160

$$\Pr(X \le x_{u_1,j} + u_1) = 1 - \Pr(X - u_1 > x_{u_1,j} \mid X > u_1) \Pr(X > u_1)$$

$$= 1 - [1 - j/(m+1)]0.5 =: q_{u_1,j},$$

with an analogous formula following for  $u_2$ , i.e.,  $q_{u_2,j} := 1 - [1 - j/(m+1)]0.99$ . The resulting probability sets  $\{q_{u_1,j}\}_{j=1}^m$  and  $\{q_{u_2,j}\}_{j=1}^m$ , with m=100, are illustrated in Figure 2. This demonstrates clearly that the lower the threshold level u, the lower the quantile probabilities evaluated by the EQD metric. Thus, quantiles lying far out into the tail of the data will carry significantly less weight for lower thresholds than for higher thresholds.

**Figure 2.** The probability sets  $\{q_{u_1,j}\}_{j=1}^m$  and  $\{q_{u_2,j}\}_{j=1}^m$  illustrated in red and black vertical lines, respectively. The left and right plots are given on different intervals to illustrate the fact the quantile probabilities exist in entirely different subregions of [0,1].

On a similar note, we remark that the metric described in equation (2) is equally weighted across all probability levels. We argue that this somewhat disagrees with intuition in the sense that many practitioners mainly care about a models' ability to estimate very extreme return levels, and one only wants observations in the tail to be driving this estimation. Including non-extreme observations will bias the estimation procedure and therefore assessing quantile discrepancies mainly for lower quantile levels, as will occur for lower candidate thresholds, provides little to no intuition as to how the fitted model will perform at the most extreme levels.

Taking these points into account, we propose an extension of the EQD procedure to improve the model fit to the most extreme observations. Our proposed extension results in models fits which more accurately capture the upper tail of the data in contexts where the EQD method struggles. Specifically, in the context of coastal modelling, we demonstrate that the EQD approach selects thresholds that do not appear appropriate for capturing the most extreme observations across many coastal sites; such issues do not arise for our extended approach. Consider the example illustrated in Figure 3 for a tide gauge record located in Penseola-Pensacola Bay, US, which is in the Gulf of Mexico. This record was selected as it is located in a region impacted by tropical cyclones, where the uncertainty in the model fits using the historical records is typically large. As demonstrated in the left panel of this figure, the model fit obtained using the EQD approach performs poorly within the upper tail. For this particular example, this indicates that the overall model fit is being driven mainly by lower observations, biasing the fit in the upper tail. Such findings were replicated across many coastal sites, indicating that this is not an unusual phenomenon. We also

illustrate the model fit that arises from our proposed method (see Section 4) in the right panel of Figure 3. One can observe that even though the updated model fit has a higher disrepency value d(u), the model quantiles appear better able to capture the upper tail in the data.

Figure 3. QQ plots for the thresholds selected using the EQD (left) and TAILS (right) approaches; see Section 4 for more details of the TAILS method. The sub captions in both cases gives the EQD score d(u) at the threshold chosen by both methods.

These findings indicate that whilst the EQD approach outperforms many existing techniques, it can, in some cases, result in model fits that fail to capture the most extreme observations. This drawback motivates novel developments, and in this work we propose an adaptation of the EQD technique, which we term the *Tail-informed threshold selectionmethodology with quantile matching for extreme value modelling* (TAILS) approachmethod. Unlike the EQD approach, our technique focuses exclusively on quantiles within a pre-defined upper tail of the data, independent of the choice of threshold. Furthermore, we demonstrate in Section 5 that TAILS results in improved model fits across a wide range of tide gauge records. Code for implementing the TAILS approach is freely available online at https://github.com/callumbarltrop/TAILS.

## 4 The TAILS approach

In this section, we introduce the TAILS approach for GPD threshold selection. To begin, let  $\mathscr{P} := \{p_i : i = 1, ..., m\}$  denote a set of increasing quantile levels close to 1: the selection of  $\mathscr{P}$  is subsequently discussed. Given a candidate threshold u, let  $x_u^b$ , b = 1, ..., B, be defined as in Section 3 and let  $\pi_u = \Pr(X \le u)$ . We propose the following metric

175 
$$\tilde{d}_{b}(u) := \frac{\sum_{i=1}^{m} \mathbb{1}(\pi_{u} 

**Figure 4.** The results from applying the EQD and TAILS methods to every GESLA record used in this study, showing the <u>distributions of</u> quantile <u>probability probabilities</u> of the selected thresholds.

obtains smaller scale parameters and larger shape parameters. This further illustrates the TAILS approach results in heavier tails in the subsequent GPD model fits.

## **5.3** Right-sided Anderson-Darling (ADr) test

The ADr test statistic (Sinclair et al., 1990; Solari et al., 2017) is used to measure the goodness-of-fit of the exceedances over the thresholds selected using both the EQD and TAILS methods, as well as the model fits computed using the Q98 approach. The test compares the theoretical quantiles against the empirical distribution, with more weight placed on the tails of the distribution (hence right-sided). The statistic quantifies the deviation of the data from the specified distribution. A *p*value\_value is obtained by bootstrapping the test statistic, with *p* indicating the probability of observing such a deviation under the null hypothesis that the threshold exceeding data cannot be modelled by follows a GPD. The null hypothesis is typically rejected for *p*values exceeding\_values below 0.05, corresponding to a 5% significance level.

A larger test statistic (equivalently, a lower *p*-value) indicates more deviation from the model distribution being tested, which in this case, is a GPD. As shown in Figure 6a, the EQD approach yields larger ADr test statistics than the TAILS method. The range of test statistics computed using the TAILS method are all less than 1, whereas the EQD approach has many values exceeding 1. This indicates the EQD method could be selecting a threshold over which the exceedances are not well characterised by a GPD. This is further corroborated by the p-values obtained for each method, plotted in Figure 6b. The median p-value across all model fits obtained using the TAILS method is 0.615, compared with 0.312 for the EQD approach. The TAILS method also outperforms the Q98 approach, with a smaller test statistic average and greater average p-value. While

**Figure 5.** Spatial plots of a) the quantile probabilities of selected thresholds using the TAILS methods, and b) the difference in the quantile probabilities of the selected thresholds between the TAILS and EQD approaches.

all the methods achieve adequate fits for most of the dataset, in some of the cases where the EQD and Q98 method lead to poor model fits (p-value less than 0.05), the TAILS method can significantly improve results. Of the 417 tide gauge records that were assessed, 89 records had an ADr p-value of less than 0.05 when using the EQD method. By comparison, using the TAILS approach, we obtain only 17 model fits with ADr p-values less than 0.05.

## 5.4 Distance metrics and return levels

As a further illustration, consider the model fits for two sites; Apalachicola in the US and Fishguard in the UK, both shown in Figure 7. The two sites have been selected based on the differences in geographic location and the associated extreme water level drivers, which lead to contrasting return level estimates. Apalachicola, located on the western coast of Florida in the Gulf of Mexico, is subjected to violent tropical cyclones, which drive huge storm surges due to the large and shallow continental shelf (Chen et al., 2008; Zachry et al., 2015). The GPD model fit that characterises the return levels of the water

**Figure 6.** Box and whisker plots showing the results from applying an ADr test to all the exceedances over the thresholds selected using the EQD and TAILS approaches, as well as using a static Q98 threshold.

level record therefore has a large positive shape parameter, which displays a steep and exponentially increasing return period curve. In contrast, Fishguard is located on the southern side of Cardigan Bay, near the inlet of the Irish Sea. The events driving extreme sea levels in this location are a combination of strong extratropical storms and astronomical tidal variation, which are characterised by a wholly different return period curve (Amin, 1982; Olbert and Hartnett, 2010). The GPD model fit for this record has a negative shape parameter, which means that the return levels plateau as the return period increases.

In the top row of Figure 7 (panels a and b), one can observe the EQD and TAILS distances metrics (i.e., equations (2) and (3)) plotted as a function of the threshold probability for both tide gauge records. Clearly, the global minimums of both approaches are starkly different, representing the different quantile estimates evaluated by either each approach. Panels c and d of Figure 7 show the estimated return levels and 95% confidence intervals from each of the TAILS, EQD and Q98 methods, at Apalachicola and Fishguard, respectively.

In the case of Apalachicola, the minimum distance (panel a) obtained using the TAILS method (0.012) is greater more than double the minimum distance obtained using the EQD approach (0.005). Compare this with the return level estimates from each of the 3 methods presented (panel c). Despite having a larger minimum distance, the TAILS approach captures the empirical observations much better than the EQD method. In fact, four of the historical events even lie outside of the 95% confidence interval for the EQD method, highlighting the need for the TAILS methodextension.

Contrast this with the results from Fishguard (panel b), where the minimum distances (panel b) obtained using each approach are much more comparable; 0.005 for TAILS and 0.004 for the EQD approach. The resulting return level estimates (panel d) are also similar, with very small differences in the mean return levels between each of the three methods. The key difference observed in panel d is the uncertainty bounds, with the EQD method having better constrained uncertainty in the higher return periods than the other two methods.

**Figure 7.** Model fits for two locations. Left column: Apalachicola, US (a and c). Right column: Fishguard, UK (b and d). The top row (a and b) shows the TAILS and EQD distance metrics, plotted as a function of the threshold probability. The vertical dashed lines indicate the distance minima, and therefore the selected threshold quantile probability. The bottom row (c and d) displays the return level plots for both methods, alongside the empirical plot and model fit obtained by using the Q98 approach. The shaded areas indicate the 95% confidence interval, calculated using bootstrapping of the GPD model-parameters.

## 6 Discussion

In this work, we have introduced an automated threshold selection technique that addresses certain limitations of the a leading existing approach. Using a global tide gauge dataset, both methods are have been rigorously compared in Section 5 alongside a commonly used static threshold. We demonstrate that in many examined spatial patterns in the model fits from the TAILS approach, along with patterns in the differences between the TAILS and EQD approaches. Furthermore, we have tested the goodness-of-fit of the resulting GPD model fits using an ADr test. Two tide gauge records were investigated in more detail to highlight the differences in the EQD and TAILS distance metrics, and to demonstrate how the parameter uncertainty changes between the different approaches.

## **6.1** Comparisons to existing approaches

At all locations, the TAILS method selects higher thresholds than the EQD approach. Particularly large increases are observed in north east Europe, as well as South Australia. The processes driving these increases are likely multifactorial. In the Baltic Sea, for example, extreme sea level events are complex phenomena, controlled by tides, antecedent meteorological conditions (that can cause prefilling of the basin), seiches and storm surges (Groll et al., 2025). The tidal range in the Baltic Sea is very small (less than 10 cm in some locations), and so any non-tidal variability in sea level is much larger relative to the daily oscillation of the sea level due to tide. This could have an impact on the EQD approach, although it is unlikely to explain all the differences. Other regions in the world also have relatively small tidal ranges, such as the Mediterranean and Gulf of Mexico, and yet these areas do not show such large increases in the quantile probabilities selected by the TAILS method compared to the EQD. As shown in Appendix A3, the length of the record and the distribution of rare, extreme observations within the record can have an impact on the threshold that is ultimately selected, although the effect tends to be muted once the record length is greater than 40 years. Other factors that could affect the selected thresholds include the meteorological forcing type (i.e. tropical cyclone vs extratropical storm) and the dominant driver of extreme water levels in a particular location (e.g. storm surge, waves or tides), but determining the relative impacts of each component remains beyond the scope of this study.

Regardless of why discrepancies occur, we demonstrate that in most cases, the TAILS approach better captures the most extreme observations compared to the EQD technique, existing EQD technique and outperforms the static Q98 threshold when assessed using an ADr test.

The TAILS method guarantees that the resulting model fits will be driven by data observed in the tail, which is desirable for practical applications where estimation of extreme quantities (e.g., return levels) is required. We also believe that calibrating threshold selection to focus on the tail will encourage more hope that these reasons, combined with the fact that automated procedures allow one to apply the POT method across a large number of locations without the need for manual checks on individual sites, will encourage practitioners to adopt our approach, since we are more likely to obtain a model fit that accurately eaptures the tail behaviour and utilise our approach.

However, focusing on model fits within the

# **6.2** Sensitivity to extreme observations and parameter uncertainty

Focusing the model fit to the upper tail comes at the cost of additional uncertainty, since by definition, since less data is available for inference. Since As uncertainty quantification is a key focus of the approach proposed by Murphy et al. (2025), the EQD technique will generally offer lower model uncertainty compared to TAILS. In some applications, this may be more desirable than capturing the most extreme observations. Thus, when deciding whether to use EQD or TAILS, one must consider the following question: is it more important that the model is more certain and robust, or that the model better captures the most extreme observations? We recommend that practitioners consider this question within the context of their application before selecting a technique.

For the application demonstrated in this paper, acknowledging and embracing uncertainty is key for any practitioner. Take the example of Apalachicola, US given in Section 5.4. This region is impacted by tropical cyclones, making the return level estimates made from the historical record very uncertain. To illustrate this point, two major Category 4 hurricanes (Helene and Milton) made landfall on the west coast of Florida in September and October 2024, after the GESLA 3.1 update was collated. Preliminary data recorded during the event suggest that Hurricane Helene broke the highest recorded water levels at three tide gauges located in Florida, and Hurricane Milton set the second highest water level ever recorded at the tide gauge located in Fort Myers, US (Powell, 2024a, b). Fitting distributions to these records pre and post these events would likely result results in different mean return levels being estimated, especially when considering the most extreme return periods (e.g., the 1 in 500 year event). We tested this and found that, when using the TAILS approach, the mean return level for the 1 in 500 year event increased by 55 cm if the tide gauge record is extended beyond the GESLA 3.1 update, to include these events. By recognising the uncertainty in the underlying processes and the uncertainty inherent in the estimates made from observations, we can be more confident that our models will be able to capture extreme events which are yet to occur.

Future work could include a variable baseline event, which is linked to the underlying forcing mechanisms in an area. As discussed in Section 5.4, tide gauges around the world are characterised by different patterns of extreme water levels. It might be possible to link a dominant forcing type to the baseline event, which could further improve the ability of TAILS to capture the tail behaviour in the estimated return levels. Another direction of future workcould be to extend the method to include non-stationary data by allowing the GPD parameters to be functions of time or covariates (e.g., Eastoe and Tawn, 2009; Youngman, 2019)

-

## **6.3** Incorporating threshold uncertainty

Our results indicate that in certain examples the Q98 approach outperforms the EQD; however, the benefits of a data-driven approach cannot be understated. When relying on TAILS or the EQD, not only is the threshold justified by a goodness-of-fit measure but sampling variability has also been taken into account. This leads to a well-justified threshold choice and an easier characterisation of the uncertainty in the resulting estimates. It also allows for the uncertainty in the threshold choice to be incorporated when making inference. As shown in Murphy et al. (2025), including this additional uncertainty results in well-calibrated confidence intervals. It should be noted that when estimating the confidence intervals for the return level estimates shown in Figure 7, we did not account for the uncertainty in the threshold itself. However, the results of a sensitivity test against record length, shown in the Appendix (Figure A4), appear to show the TAILS approach leads to lower threshold uncertainty. This is encouraging and might help in the trade-off with additional parameter uncertainty that is introduced using the TAILS method. Overall, including this uncertainty may improve our method's ability to capture unobserved extreme events and could provide a better understanding of the uncertainty in return level estimates beyond the observed data. This remains a further avenue of research for our framework.

# 6.4 Incorporating more complex characteristics within the TAILS approach

Throughout this work, we make the implicit assumption that data are identically distributed, even though we acknowledge that environmental processes such as sea levels are unlikely to be stationary in an ever-changing climate. This choice was motivated by practical implications; stationary models are simpler to implement and best practices (i.e., using a POT model) are well established. Moreover, when applying simple stationary models to such contexts, the TAILS approach may be favoured as the generally higher threshold choices should help remove the influence of some covariate effects, leading to more stationary time series. However, there is no reason why one could not incorporate covariate dependence into the threshold and parameters of a GPD (e.g., Davison and Smith, 1990; Chavez-Demoulin and Davison, 2005) when applying the TAILS or EQD approach. Failure to incorporate covariate effects may help to explain the reason for the poor fit in the upper tail in e.g. Figure 3 when using the EQD. Accounting for this aspect in the EQD or TAILS approach could allow for the use of lower thresholds without the loss of accuracy for more extreme observations, providing a way to balance between the two goals mentioned above, i.e., uncertainty and accuracy in the upper-tail.

A wide range of modelling approaches have been proposed for incorporating covariate effects into POT modelling (e.g., Eastoe and Tawn Relevant covariates are those that impact the number of extreme events that occur within a given year; for example, indices related to the ENSO and NAO phenomena, which affect the likelihood of temperature and precipitation extremes (Dong et al., 2019), can be incorporated into the POT modelling framework. Continuing to develop could be incorporated when specifying a model for sea level data. Only minor modification would be needed to apply the TAILS or EQD approaches here; specifically, we would assess quantile discrepancies on a transformed scale, rather than the observed scale (see Varty et al. (2021) for related discussion). However, we note that standard practices for applying non-stationary POT models are not well established, and it is not clear how one should select which covariates to include, or how flexible a model is required. The development of automated threshold selection approaches to suit a wide range of different for non-stationary data structures represents an important line of future research.

We also remark that we assume a constant baseline event for our approach. Future work could incorporate a variable baseline event linked to the underlying forcing mechanisms in an area. As discussed in Section 5.4, tide gauges around the world are characterised by different patterns of extreme water levels. It might be possible to link a dominant forcing type to the baseline event, which could improve further the performance of the TAILS approach.

While results may indicate in certain examples that the Q98 approach outperforms the EQD, the benefits of a data-driven approachean not be understated. When relying on TAILS or the EQD, not only is the threshold justified by a goodness-of-fit measure but sampling variability has also been taken into account. This leads to a well-justified threshold choice and an easier characterisation of the uncertainty in resulting estimates. It also allows for the uncertainty in the threshold choice to be incorporated when making inference; see Murphy et al. (2025). Furthermore, when applying methods to a large number of sites, employing an automated procedure avoids the need for manual checks on individual threshold choices Finally, we acknowledge that our automated selection technique could be useful for improved threshold estimation in the wider context of multivariate and spatial extremes. With suitable adjustment, the TAILS technique could help when implementing approaches employing the multivariate regular variation framework (e.g., Tawn, 1990; de Carvalho and Davison, 2014; Padoan et al., 2010), or alternative frameworks for variables exhibiting asymptotic independence (e.g., Ledford and Tawn, 1996; Heffernan and Tawn, 2004; Wadsworth and Tawn, 2004; Wa

. The data-driven approach would allow for the threshold estimation uncertainty to be propagated through to joint tail inferences.

The development of automated threshold selection approaches in multivariate and spatial settings has been largely overlooked in the literature, thus representing an natural avenue for future work.

Finally, we note that the selection of

# **6.5** Selecting tuning parameters

TAILS requires a selection of several non-trivial tuning parameters; this includes the probability set Pis non-trivial, as discussed, m, and the limit on candidate thresholds, which we define as the 1 year return level in Section 4. We therefore Our choices were motivated by the specific application at hand, and we consequently recommend that practitioners experiment with both the baseline and maximal probabilities these parameters to assess whether such values have a practical effect on the resulting model, using diagnostics such as QQ and return level plots to guide this procedure. The code has been written in such a way as to make it easily parallelised, allowing for fast testing of multiple baseline and maximal probabilities across a variety of datasets. We encourage and invite fellow researchers to utilise this method on other perils, such as rainfall or river flow measurements. Exploring data-driven techniques (e.g., cross validation) for selecting tuning parameters of automated threshold selection approaches remains an open area for novel developments.

## 7 Conclusions

Accurately estimating the extreme tail behaviour of historical observations is of great importance to researchers and practitioners working in natural hazards. POT methods are regularly used in these fields for this purpose, but selecting the threshold above which to consider an exceedance requires careful consideration. In this paper, we present TAILS, a new method for automating the threshold selection process building upon the recently published EQD method (Murphy et al., 2025).

We apply two key innovations to improve upon the EQD method in the context of extreme coastal sea levels. Firstly, we fix the quantiles that we consider when computing the distance metrics. This avoids oversampling the most extreme quantiles when assessing higher thresholds. Secondly, we limit the quantiles considered for our distance metric to be only above a predetermined baseline probability. This means that when optimising the distance metric to select a threshold, we are only considering quantiles that we deem to be extreme, and hence worth considering when selecting a threshold. In this study, the baseline probability was decided using the literature and a sensitivity test.

We show that the TAILS approach selects, on average, higher thresholds than the EQD method. When the resulting model fits are evaluated using an ADr test against the EQD method and the Q98 method approaches, we show that the TAILS method outperforms both with respect to the ADr test statistic and the p-value. We also illustrate that the TAILS method typically results in larger uncertainty bounds, but argue that this is not necessarily a negative when considering water level records located in regions that experience tropical cyclones, this is positive highly variable regions experiencing tropical cyclones.

Although a large number of records are assessed, this study is limited in scope as it only considers tide gauge recordsdata. We hope that the method can be widely used to better estimate the intensities improve the estimation of magnitudes and

frequencies of other natural hazards. The code has been written in such a way as to make it easily accessible and easily parallelised interpretable so as to encourage uptake from fellow researchers. We believe we have clearly demonstrated the potential of the TAILS approach, alongside its advantages compared to existing techniques.

Code and data availability. The code for implementing the TAILS approach is freely available online at https://github.com/callumbarltrop/TAILS, along with an example data set. The GESLA 3.1 tide gauge database is available from the corresponding author upon reasonable request.

# Appendix A: Sensitivity test of baseline probability, $p_1$ tests and supporting figures

# A1 Sensitivity test of baseline probability, $p_1$

A range of baseline probabilities were tested across the whole dataset, and the resulting threshold and model fits were used to calculate a right-sided Anderson-Darling (ADr) test statistic and the p-value (Sinclair et al., 1990; Solari et al., 2017). For more details on the ADr test, see the main text. The return periods that were tested for the baseline probabilities were 0.083, 0.167, 0.25, 0.33, 0.5, 0.667, and 1.0 years. These equivalate equate to the 1 in 1, 2, 3, 4, 6, 8 and 12 month events.

The results of this sensitivity test are shown in Figure A1. Panel a) presents the ADr test statistic for the 7 return periods tested. When looking at the median and interquartile ranges of the ADr test statistics, the threshold selection looks relatively insensitive to the return period chosen, with very little differences between the 0.167, 0.25, 0.333, and 0.5 year return periods. When considering the ADr test p-value (in panel b), there is also only small differences between the 0.167, 0.25, 0.33 and 0.5 year return periods. We take this, along with the value obtain obtained from the literature (presented in main textarticle), as evidence that any one of these values would suffice as the baseline probability,  $p_1$ .

**Figure A1.** The results from the sensitivity test of different baseline probabilities.

# 445 Appendix B: Sensitivity test of number of quantile levels, m

## A1 Sensitivity test of number of quantile levels, m

Following Murphy et al. (2025), a sensitivity test to the number of quantile levels, m was carried out. The values of m tested were 10, 50, 100, 200, 500, 1000 and 'n\_exceedances', which is equal to the number of exceedances over the baseline probability for each tide gauge record. The range of m values that are used by the 'n\_exceedances' are shown below in Figure A1. The full range spreads between 161 to 811, and the median is centred on 231.

Figure A1. The range of m values used by 'n\_exceedances', which is equal to the number of exceedances over the baseline probability for each tide gauge record.

The results of this sensitivity analysis are presented in Figure A2, showing that the method is quite insensitive to the m value used. This is similar to the findings of Murphy et al. (2025). We recommend using any value over 10, and choose to use m = 500 in this study for consistency with Murphy et al. (2025).

Figure A2. The results of the sensitivity test using different m values. 'n\_exceedances' refers to the number of exceedances over the baseline probability, at each tide gauge record.

# A2 Spatial patterns in the scale and shape parameters of the GPD

In addition to the spatial plot of the quantile probabilities of selected thresholds, shown in Figure 5, we present here the spatial patterns of the scale and shape parameters of the GPD. The scale parameter obtained using the TAILS method is shown in Figure A3a and the shape parameter is shown in Figure A3c. The differences between the scale and shape parameters obtained using the TAILS method vs the EQD approach are shown in Figures A3b and d, respectively.

**Figure A3.** Spatial plots of the scale parameter (a) and the shape parameter (c) of the GPD when using the TAILS method. The difference in scale parameter obtained using the TAILS approach vs the EQD approach is shown in panel b and the difference in the shape parameter is shown in panel d.

The scale parameters are generally quite small, with the exception of German/Danish coastlines, which have values around 0.3 - 0.4. Overall, we see a reduction in the scale parameter obtained using the TAILS approach when compared with the EQD. Some locations show increases, such as along the German/Danish coast, Japan and North East US. The shape parameters have more variability globally, with strong positive values present along the US east coast and Caribbean. Europe generally exhibits negative shape parameters, which are more common for areas impacted by extratropical storms, although some outliers persist. When comparing the differences between the TAILS and EQD approaches, we see increases in the shape parameter in the vast majority of locations. This supports the observation that using the TAILS approach results in heavier tail estimates for the GPD.

# A3 Sensitivity test to the length of record and number of extreme observations

A test was carried out to determine the sensitivity of the TAILS method to the length of the record and the number of extremes present in the record. Whole years from the tide gauge records investigated in Section 5.4 (Apalachicola and Fishguard) are randomly sampled using 200 bootstraps with replacement to create synthetic records of length 10, 20, 30, 40, 50, 60, 70 and 80 years. The records are then declustered using a 4-day storm window, and the TAILS threshold is obtained. Comparisons are made against the EQD method for reference. The number of extreme events in each bootstrapped sample is obtained. An extreme event is defined as a water level in the bootstrapped sampled record that is greater than the 0.99 quantile of the original declustered record. The distributions of the quantile probabilities of the TAILS and EQD selected threshold are plotted as box plots below, in Figures A4a, b, c and d. Figures A4e and f show the number of extreme events plotted against record length, with the colour map of the markers illustrating the quantile probability of the TAILS threshold.

Figure A4. Sensitivity test of record length and the number of extreme events present in the record against the quantile probabilities of the selected thresholds. The left column shows the results for Apalachicola, US, and the right column is Fishguard, UK. Panels a and b compare record length against the EQD method. Panels c and d compare record length against the TAILS method. Panels e and f show the results comparing record length against the number of extreme events present in the records, along with the quantile probabilities of the selected thresholds using the TAILS approach.

The results of the test show that the TAILS method is generally insensitive to the record length, once records are greater than 30-40 years. Short records of less than 20 years tend to have a lower selected threshold compared with longer records. Shorter records also have greater variability, but this reduces as the record length increases. This is in contrast to the EQD approach, which generally has greater variability, regardless of record length. At Fishguard, the record length makes very little difference to the selected threshold when using the EQD approach. In panels e and f, record length, number of extreme events and the quantile probabilities of the selected thresholds using the TAILS method are assessed. These results show that there is no clear trend between the number of extreme events present in a record and the threshold selected.

Author contributions. TPC was responsible for making the edits to the original code, preparing the data for use in the study, validating the results and drafting the introduction, data, results, discussion and conclusion sections of the manuscript, as well as reviewing the manuscript. CJRMB provided guidance and expertise in interpreting the results, helped with ideas on how to improve the original method, and contributed to the manuscript, including writing parts of the abstract, POT modelling, the TAILS approach and discussion sections, as well as carrying out thorough reviews. CM kindly provided the original code that underlies the EQD method, and has contributed to the manuscript by writing parts of the introduction, POT modelling and the TAILS approach and discussion sections, whilst also helping with thorough reviews of the manuscript throughout. IDH provided the GESLA 3.1 update, initial guidance, and ideas about how to start, as well as reviewing the manuscript. PDB provided guidance and also reviewed the manuscript. NDQ provided guidance, advice, and support throughout the study, offering insight in interpreting the results and ideas on how to proceed, as well as reviewing the manuscript.

Competing interests. The contact author has declared that none of the authors have any competing interests.

TEXT-

480

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
