# Peer review of "Automated tail-informed threshold selection for extreme coastal sea levels"

_EGUsphere, 2025_

## Author Comment (AC2)

**Response to review of 'Automated tail-informed threshold selection for extreme coastal sea levels'**

Thomas P. Collings[1], Callum J. R. Murphy-Barltrop[2,3], Conor Murphy[4], Ivan D. Haigh[1,5], Paul D. Bates[1,6], and Niall D. Quinn[1]

[1]Fathom, Floor 2, Clifton Heights, Clifton, Bristol BS8 1EJ, UK
[2]Technische Universität Dresden, Institut Für Mathematische Stochastik, Dresden, Germany
[3]Center for Scalable Data Analytics and Artificial Intelligence (ScaDS.AI), Dresden/Leipzig, Germany
[4]School of Mathematical Sciences, Lancaster University, Lancaster, LA1 4YF, UK
[5]School of Ocean and Earth Science, University of Southampton, National Oceanography Centre, European Way, Southampton SO14 3ZH, UK
[6]School of Geographical Sciences, University of Bristol, Bristol BS8 1SS, UK

**Correspondence:** Thomas P. Collings (t.collings@fathom.global)

We would like to thank both reviewers for providing detailed and constructive feedback. The comments provided have helped to improve the quality and readability of our article. In the below text, we respond to each comment in turn, highlighting any novel additions in the text.

**Reviewer 1**

5   This study proposes a novel automated threshold selection method for modeling extreme coastal sea levels within the Peaks Over Threshold (POT) framework, aiming to better capture tail behavior while addressing the limitations of arbitrary and existing automated threshold choices. The method is applied to global tide gauge data and evaluated using the Anderson-Darling test, demonstrating improved performance over conventional techniques. However, the quality and readability of Figure 4 should be enhanced to ensure clearer communication of results. The discussion section is relatively weak, lacking depth, 10  logical structure, and clarity. It is recommended that the discussion be expanded and subdivided to include specific commentary on the data, methodology, and results of this study, with comparative insights drawn from previous research to highlight the strengths and limitations of the proposed approach. The authors are also encouraged to include a forward-looking perspective outlining directions for future work. In summary, I recommend a major revision.

We thank the reviewer for this useful feedback. We have made several additions to the article to account for these comments:

15    1. We have removed Figure 4 and replaced it with an alternative histogram figure (see below), alongside detailed spatial figures illustrating the increases in selected threshold levels across different locations. This figure makes it clearer how the use of the TAILS approach always results in high threshold choices compared to the EQD technique. Furthermore, we have also provided a more in-depth spatial analysis of observation sites, showing GPD parameter estimates and return levels obtained using the TAILS method. Such plots illustrate some spatial patterns within the model fits across 20     sites; however, due to the sparsity of sampling locations, we can only clearly identify patterns in regions with large

numbers of observation stations. These figures are shown in the response to reviewer 2.

[Figure]

**Figure 1.** Updated Figure 4 in the manuscript. The results from applying the EQD and TAILS methods to every GESLA record used in this study, showing the distributions of quantile probabilities of the selected thresholds.

25  2. Following this recommendation, we have greatly expanded the discussion and split sections 5 and 6 into multiple subsections. We provide a more detailed breakdown of the results and conclusions from our study. We also highlight several areas of future work, including accounting for threshold uncertainty and expanding the TAILS approach to account for non-stationary and multivariate data. Moreover, we better highlight the relative strengths and weaknesses of TAILS compared to existing approaches.

30  **Reviewer 2**

The manuscript entitled *Automated tail-informed threshold selection for extreme coastal sea levels* by Collings et al. presents a methodological advancement for improving the threshold selection process in the Peaks-Over-Threshold (POT) framework. The authors propose an automated, data-driven approach tailored for extreme coastal sea level analysis, addressing a key bottleneck in both scientific and operational applications of extreme value theory.

35  The proposed method is potentially useful for practitioners, particularly in settings where arbitrary or fixed thresholds are problematic and require substantial expertise and technical judgment from the user. The manuscript is timely and relevant for NHESS, as it contributes to a sensitive topic and to the ongoing discussions around robust quantification of coastal hazards. It is generally well-written and understandable.

However, the discussion of the results should be expanded and more clearly structured to enhance the scientific impact of the work. The validation of the method—although supported by two illustrative case studies—remains somewhat limited. A more comprehensive evaluation of the model's performance across a wider range of sites or under varying data availability conditions would help assess its robustness and provide stronger support for broader applicability.

For these reasons, I recommend a major revision before the manuscript can be considered for publication.

We thank the reviewer for this useful feedback. As mentioned above, we have greatly expanded our discussion of the results, splitting sections 5 and 6 into multiple in-depth subsections. We also considered applying the approach to a wider range of sites, but decided against this for several reasons. Firstly, we believe our case study is already comprehensive, covering a significant area of the world with 417 study locations. Finding additional data sets of comparable quality and study length is also not straightforward and would represent a significant research project in itself without any guarantee of success. We therefore opted to instead vary the data availability using the GESLA database, and assess how this impacts the performance of TAILS. We provide detailed responses to the provided comments below.

**MAJOR COMMENTS:**

1. It would be useful to explore TAILS approach sensitivity to the length of the calibration sample, i.e., the number of available years of observation. Since the method is appealing for practical applications by professionals and practitioners, understanding its robustness under limited data availability would greatly increase its usability in poorly gauged sites;

   Given the focus of our approach is on adjusting the EQD approach to focus on tail observations, we would naturally expect a reduction in performance for lower sample sizes, since this in turn reduces the number of observed extreme events. To test the sensitivity in this case, we applied the TAILS approach to a subset of sites and randomly varied the size of the observation period. These results are presented in the appendix. Encouragingly, even though the tuning parameters were selected using the full sample, the chosen GPD thresholds and parameters do not change significantly when the sample size is reduced, suggesting our approach can be robust to the size of the observation period. The same could not be said for the EQD method, which varied more with the sample size. Of course, this trend would not continue indefinitely, and one should always aim, when possible, to use data-rich samples when applying extreme value modelling techniques.

[Figure]

**Figure 2.** New Figure A5 in the appendix. Sensitivity test of record length and the number of extreme events present in the record against the quantile probabilities of the selected thresholds. The left column shows the results for Apalachicola, US, and the right column is Fishguard, UK. Panels a) and b) compare record length against the EQD method. Panels c) and d) compare record length against the TAILS method.

Panels e) and f) show the results comparing record length against the number of extreme events present in the records, along with the quantile probabilities of the selected thresholds using the TAILS approach.

2. Given the more weight attributed to the tail of the events population, that is the novelty and authors contibution to overcome existing approaches limitations, the more extreme events presence in data records effects are even more important in the proposed technique. I am wondering how sensitive the model threshold selection is to very rare events compared to the other methods. Artificially removing/adding adding large enough events or, consistently with previous point, years containing large enough events, and testing obtained thresold/quantiles can give some useful insights; In line with the previous comment, we applied the TAILS approach across a subset of sites while randomly varying the size of the observation period. This has the same effect as randomly removing and/or including certain extreme events from the time series, thus allowing us to assess the sensitivity to certain events. As expected, there was some discrepancy as we altered the size of the observation window, but this was not significant, suggesting our approach appears relatively robust to one-off extreme events. The results of this are shown in panels e) and f) of the figure above. In this test, an extreme event is defined as a water level in the bootstrapped sampled record that is greater than the 0.99 quantile of the original declustered record.

3. Considering the global scale of the analyses conducted, it would be valuable to assess whether any spatial patterns emerge in the performance of TAILS relative to conventional methods. Identifying systematic spatial behaviours, if any, could offer useful insights for practitioners and strengthen the case for its broader adoption, especially where spatially consistent behavior is observed; We have added a range of spatial plots (see below) illustrating several features of the model fits, including the quantile probabilities of the selected thresholds from the TAILS approach, and the change in threshold from the EQD to the TAILS method. This figure has been added into the manuscript as Figure 5. We have also included spatial plots of the shape and scale parameters of the GPD, as well as the differences between the EQD and TAILS which has been added to the appendix. All thresholds selected using the TAILS method are greater than the thresholds selected by the EQD. Strong spatial patterns are present particularly at tide gauge locations in north-eastern Europe. The tide gauge records with the largest increases are located in the Baltics, showing changes of nearly 0.5. Spatial trends are also visible around Australia, with the TAILS approach selecting higher threshold probabilities around the south of the country compared with the north. Similar patterns appear evident for northern Japan and the north-west US. The scale parameters are generally quite small, with the exception of German/Danish coastlines, which have values around 0.3 - 0.4. Overall, we see a reduction in the scale parameter obtained using the TAILS approach when compared with the EQD. Some locations show increases, such as along the German/Danish coast, Japan and North East US. The shape parameters have more variability globally, with strong positive values present along the US east coast and Caribbean. Europe generally exhibits negative shape parameters, which are more common for areas impacted by extratropical storms, although some outliers persist. When comparing the differences between the TAILS and EQD approaches, we see increases in the shape parameter in the vast majority of locations. This supports the observation that using the TAILS approach results in heavier tail estimates of the GPD. We acknowledge that the sparsity of observation stations makes it difficult to identify clear patterns in many regions, and as such, all identified patterns tend to occur around Europe, Japan, USA or Australia.

[Figure]

**Figure 3.** New Figure 5 in the manuscript. Spatial plots of a) the quantile probabilities of selected thresholds using the TAILS methods, and b) the difference in the quantile probabilities of the selected thresholds between the TAILS and EQD approaches.

[Figure]

**Figure 4.** New Figure A4 in the appendix. Spatial plots of the scale parameter (a) and the shape parameter (c) of the GPD when using the TAILS method. The difference in scale parameter obtained using the TAILS approach vs the EQD approach is shown in panel b) and the difference in the shape parameter is shown in panel d).

4. The study provides a good validation of the proposed method based on the exceedences over the thresholds goodness-of-fit. Given the practical relevance of the proposed method, I would suggest to include a more comprehensive benchmark of the model performance on the annual maxima, in addition to the two case studies currently presented. Applying the method to a larger, eventually selected, set of stations could offer a more comprehensive assessment of its return levels predictive potential, that is of primary interest from an engineering perspective.

We thank the reviewer for this suggestion, but respectfully argue from a statistical perspective that the use of peaks-over-threshold (POT) is more appropriate and widely accepted in modern extreme value analysis than annual maxima (AM). This selection is particularly relevant when the goal is to make efficient use of the available data. In our case, we have some records with only 40 years of observations available, which would give just 40 observations for the AM model. Quantities computed from this model, such as return levels, would have far higher variability compared to those obtained using POT techniques, reducing their usability and reliability.

This has been a discussion point in many significant works; for example, Davison and Smith (1990), Coles (2001) and Scarrott and MacDonald (2012) all advocate for the use of POT over AM on the basis of data efficiency and estimation accuracy. Therefore, we believe comparing our approach with AM would not add meaningful value to our study and could potentially dilute the clarity of our methodological focus.

We have updated the discussion in Section 1, page 2 to account for this comment.

**MINOR COMMENTS:**

1. I would also mention the work from Tancredi et al. (2006) in the POT modelling section as a Bayesian study that explore how to integrate uncertainty in the threshold selection; We have updated the literature review in Section 3 (page 4) to include this additional reference.

2. The authors mention the GESLA 3.1 update as a minor revision of the GESLA 3 dataset, which is provided by one of the authors. As far as I am aware, this updated dataset is not yet publicly available. The authors are encouraged to clarify whether they plan to release it, to ensure reproducibility and broader adoption of the proposed methodology. We have updated the data availability statement to clarify that the revised GESLA 3.1 database is available from the corresponding author upon request. We have been told the revised GESLA 3.1 database will be released publicly in the next couple of weeks as well.

**References**

130   Coles, S.: An Introduction to Statistical Modeling of Extreme Values, Springer London, ISBN 978-1-84996-874-4, https://doi.org/10.1007/978-1-4471-3675-0, 2001.

Davison, A. C. and Smith, R. L.: Models for Exceedances Over High Thresholds, Journal of the Royal Statistical Society. Series B: Statistical Methodology, 52, 393–425, https://doi.org/10.1111/j.2517-6161.1990.tb01796.x, 1990.

Scarrott, C. and MacDonald, A.: A review of extreme value threshold estimation and uncertainty quantification, Revstat Statistical Journal,

135   10, 33–60, 2012.

Tancredi, A., Anderson, C., and O'Hagan, A.: Accounting for threshold uncertainty in extreme value estimation, Extremes, 9, 87–106, 2006.

---

## Referee Report (RR1)

The paper "Automated tail-informed threshold selection for extreme coastal sea levels" by Collings et al. presents a new methodology (called TAILS) to define thresholds when performing Extreme Value Analysis to Peak Over Threshold data. The topic of the research aligns well with the objectives of the journal, the paper is well written and results are in general properly presented and discussed. However, there are some aspects that should be better clarified before the paper is suitable for publication. See the comments below.

First and most important, Authors use the p-value of the ADr test to prove that the TAILS method works better than two other methods, i.e., the *expected quantile discrepancy* and the Q98. It follows that the ultimate goal is to select subsets that are well modeled by a GPD; I therefore believe that other tests should have been employed for a comparative analysis, namely those sharing the same objective - see for instance the work by Solari et al., 2017, which is also mentioned in the paper. On the other hand, the use of a fixed quantile (Q98) seems more like the first step for declustering the exceedances within a two-step selection (e.g., Bernardara et al., 2012¹), which should lay the ground for a subsequent selection of a statistical threshold - therefore aiming to ensure a proper distribution fit. This should be at least commented in the paper.

Second, I found some parts of the methods hard to follow. I am not familiar with the work by Murphy et al., (2025), so I apologize in advance if some questions may look naive. At page 5, line 111, n exceedances  $x_u$  are considered. As such, shouldn't there be as many associated probabilities P? In other words, isn't m equal to n? By looking at lines 118-119 it seems so (i.e., Q is associated to  $x_u^b$ ). If that is the case, does it make sense to use a fixed value of m to compute the probabilities q for increasing thresholds? (see the last equation at page 5, which by the way should be numbered). Moving to the TAILS approach, I do not understand why in Equation (3), at the numerator, there are both  $p_i$  and  $p_j$ . Is that a typo? Moreover, if  $x_u^b$  is defined as in Section 3, that implies that it is based on a sample of excesses of a candidate threshold u. If so, I do not understand what do probabilities  $\pi_u$  represent (clearly not the chance that an excess of a threshold is lower than the threshold itself). In summary, I think that the whole methodology should be better explained to avoid any confusions.

Finally, at page 3, line 66, Authors claim that SLR is accounted for by removing a linear trend to all data. Is this a reliable assumption on a global scale? Given the length of the time series, would not an exponential trend be better suited for this purpose at least to some locations? Could you elaborate on this point in the text?

**See below other minor comments:**

- In Figure 1, the two test sites are hard to see. Consider adding an inset with a close-up on the area of interest;
- In Figure 5, using fewer colors would help appreciate the spatial differences in the results;
- At page 11, I like the use of the p-value as a GOF measure. In this respect, Authors may want to cite the seminal work by Wasserstein & Lazar (2016)2;
- Panels c and d in Figure 7 are very busy and hard to interpret;

<sup>1 Bernardara, P., Mazas, F., Weiss, J., Andreewsky, M., Kergadallan, X., Benoît, M., & Hamm, L. (2012). On the two step threshold selection for over-threshold modelling. *Coastal Engineering Proceedings*, 1(33), 1-6.

<sup>2 Wasserstein, R. L., & Lazar, N. A. (2016). The ASA statement on p-values: context, process, and purpose. *The American Statistician*, *70*(2), 129-133.

---

## Author Response (AR2)

**Response to review of 'Automated tail-informed threshold selection for extreme coastal sea levels'**

Thomas P. Collings1, Callum J. R. Murphy-Barltrop2,3, Conor Murphy4, Ivan D. Haigh1,5, Paul D. Bates1,6, and Niall D. Quinn1

**Correspondence:** Thomas P. Collings (t.collings@fathom.global)

We would like to thank the reviewer for their detailed and constructive feedback. The comments provided have helped to improve the quality and readability of our article. In the below text, we respond to each comment in turn, highlighting any novel additions in the text.

**Reviewer 3**

15

- The paper *Automated tail-informed threshold selection for extreme coastal sea levels* by Collings et al. presents a new methodology (called TAILS) to define thresholds when performing Extreme Value Analysis to Peak Over Threshold data. The topic of the research aligns well with the objectives of the journal, the paper is well written and results are in general properly presented and discussed. However, there are some aspects that should be better clarified before the paper is suitable for publication. See the comments below.
- 10 We provide detailed responses to the provided comments below.

**MAJOR COMMENTS:**

1. First and most important, Authors use the p-value of the ADr test to prove that the TAILS method works better than two other methods, i.e., the expected quantile discrepancy and the Q98. It follows that the ultimate goal is to select subsets that are well modeled by a GPD; I therefore believe that other tests should have been employed for a comparative analysis, namely those sharing the same objective - see for instance the work by Solari et al., 2017, which is also mentioned in the paper. On the other hand, the use of a fixed quantile (Q98) seems more like the first step for declustering the exceedances within a two-step selection (e.g., Bernardara et al., 2012), which should lay the ground for a subsequent selection of a statistical threshold - therefore aiming to ensure a proper distribution fit. This should be at least commented in the paper. We have selected the ADr test to assess goodness-of-fit (GOF) as it is particularly sensitive to discrepancies in the upper

<sup>1Fathom, Floor 2, Clifton Heights, Clifton, Bristol BS8 1EJ, UK

<sup>2Technische Universität Dresden, Institut Für Mathematische Stochastik, Dresden, Germany

<sup>3Center for Scalable Data Analytics and Artificial Intelligence (ScaDS,AI), Dresden/Leipzig, Germany

<sup>4School of Mathematical Sciences, Lancaster University, Lancaster, LA1 4YF, UK

<sup>5School of Ocean and Earth Science, University of Southampton, National Oceanography Centre, European Way, Southampton SO14 3ZH, UK

<sup>6School of Geographical Sciences, University of Bristol, Bristol BS8 1SS, UK

tail, which is what we aim to characterise well with our method. Furthermore, it is one of the most commonly used GOF measures from the literature (Heo et al., 2013; Gharib et al., 2017; Benito et al., 2023). In Solari et al. (2017), they use the ADr p-value as the metric by which to determine the best threshold, but we cannot find any references in the paper to other GOF measures used to assess the resulting GPD fits. The aim of this study was not to be an exhaustive review of GOF measures, but to concisely demonstrate the efficacy of our method against other commonly used methods. Using a single metric that is well suited to the area of the distribution we aim to characterise makes the comparison easier to communicate, without the added ambiguity and complexities of extra GOF measures. We therefore respectfully maintain our choice of the ADr test as the sole comparative metric, while acknowledging that other tests may also provide useful complementary perspectives in future work. As for the use of the Q98 as the first step in a two-step selection process, it is also used as the final threshold in other global studies, as referenced in the manuscript. For the avoidance of doubt, we have added the following sentence to the introduction - 'Note that the use of static thresholds, such as the Q98, are common in two-step threshold selection processes (e.g., Bernardara et al., 2012), and should not be confused with the use of the static threshold as the threshold above which to consider an exceedance.'

20

25

30

35

40

45

50

2. Second, I found some parts of the methods hard to follow. I am not familiar with the work by Murphy et al., (2025), so I apologize in advance if some questions may look naive. At page 5, line 111, n exceedances xu are considered. As such, shouldn't there be as many associated probabilities P? In other words, isn't m equal to n? By looking at lines 118-119 it seems so (i.e., O is associated to xu b). If that is the case, does it make sense to use a fixed value of m to compute the probabilities q for increasing thresholds? (see the last equation at page 5, which by the way should be numbered). Moving to the TAILS approach, I do not understand why in Equation (3), at the numerator, there are both pi and pj. Is that a typo? Moreover, if xu b is defined as in Section 3, that implies that it is based on a sample of excesses of a candidate threshold u. If so, I do not understand what do probabilities  $\pi u$  represent (clearly not the chance that an excess of a threshold is lower than the threshold itself). In summary, I think that the whole methodology should be better explained to avoid any confusions. We appreciate that the content referenced here is rather technical in nature and may be confusing at first, especially to those unfamiliar with such techniques. We have therefore added some additional comments to clarify the approaches discussed. Firstly, for each threshold u, there will be  $n_u$  exceedances, with  $n_u$  varying over u and the sample in question. However, the set  $\mathcal{P}_m$  (and value m) is defined independently of the threshold choice/sample, and as such the same probabilities are considered for every threshold. In the case when  $m > n_u$ , the quantile function Q linearly interpolates between the observed quantiles. We have adjusted the text on page 5 (now line  $\sim 115$ ) to make it clear  $\mathcal{P}_m$ is fixed over threshold and  $n_u$  is sample + threshold dependent. In practice, we do exactly as you suggested and keep mfixed. We also have labelled the final equations on page 5 (now on page 6)

In equation 3 (now equation 5), we originally opted for different indices in the numerator and denominator to make it clear these sums are computed separately before their ratio is evaluated (i.e., we don't take the sum of ratios, but rather the ratio of the sums). However, to avoid confusion, we have updated the indices to both be i.

Finally, the probabilities  $\pi_u$  represent (empirical) non-exceedance probabilities for each threshold u. We are only able to evaluate quantiles at probabilities great than  $\pi_u$ , since the GPD is obviously not valid below the threshold u. This explains the rather dense and complicated equation 3 (now equation 5), and we have a comment on page 8, line  $\sim$ 175, stating that this equation 'accounts for cases when the threshold probability,  $\pi_u$ , exceeds a subset of  $\mathcal{P}$ '. We have also added the comment 'In other words, this ensures the fitted GPD is only evaluated above the candidate threshold' to better explain the proposed method.

3. Finally, at page 3, line 66, Authors claim that SLR is accounted for by removing a linear trend to all data. Is this a reliable assumption on a global scale? Given the length of the time series, would not an exponential trend be better suited for this purpose at least to some locations? Could you elaborate on this point in the text?; Whilst SLR can be modelled as an quadratic trend, modelling it as a linear trend is common in other regional and global studies (Sweet et al., 2020; Frau et al., 2018). We tested this a sample of sites globally and show that difference between an quadratic trend and a linear trend is small (see figure below for Newlyn, UK). We have added the following sentence to section 2 - "Although some tide gauge stations show an accelerating sea level change, a linear trend is judged to be sufficient to model sea level change in this study."

**MINOR COMMENTS:**

55

60

65

70

1. In Figure 1, the two test sites are hard to see. Consider adding an inset with a close-up on the area of interest; Thank you for your suggestion. We have added two insets to improve clarity of the figure. Please see the updated figure below.

75

**Figure 1.** New Figure 1 in the manuscript, showing the locations of the GESLA records and the highlight Fishguard and Apalachicola using map insets.

2. In Figure 5, using fewer colors would help appreciate the spatial differences in the results; We have reduced the number of colours in this plot, as well as in Figure A4 which shows similar global plots. The updated Figure 5 is shown below.

Figure 2. New Figure 5 in the manuscript, with 10 colours used in each colour map.

- 3. At page 11, I like the use of the p-value as a GOF measure. In this respect, Authors may want to cite the seminal work by Wasserstein & Lazar (2016); Thank you for the information. We have added the reference to the manuscript on page 11.
- 4. Panels c and d in Figure 7 are very busy and hard to interpret; We have removed the shading of the confidence intervals as we believe this improves the clarity of the figure. Please see the updated figure below.

85

80

Figure 3. New Figure 7 in the manuscript, with the shading of the CI removed.

**References**

90

100

- Benito, S., López-Martín, C., and Navarro, M. A.: Assessing the importance of the choice threshold in quantifying market risk under the POT approach (EVT), Risk Management, 25, 1743–4637, https://doi.org/10.1057/s41283-022-00106-w, 2023.
- Bernardara, P., Mazas, F., Weiss, J., Andreewsky, M., Kergadallan, X., Benoît, M., and Hamm, L.: On the two step threshold selection for over-threshold modelling, Coastal Engineering Proceedings, 1, management.42, https://doi.org/10.9753/icce.v33.management.42, 2012.
- Frau, R., Andreewsky, M., and Bernardara, P.: The use of historical information for regional frequency analysis of extreme skew surge, Natural Hazards and Earth System Sciences, 18, 949–962, https://doi.org/10.5194/nhess-18-949-2018, 2018.
- 95 Gharib, A., Davies, E. G. R., Goss, G. G., and Faramarzi, M.: Assessment of the Combined Effects of Threshold Selection and Parameter Estimation of Generalized Pareto Distribution with Applications to Flood Frequency Analysis, Water, 9, https://doi.org/10.3390/w9090692, 2017.
  - Heo, J.-H., Shin, H., Nam, W., Om, J., and Jeong, C.: Approximation of modified Anderson–Darling test statistics for extreme value distributions with unknown shape parameter, Journal of Hydrology, 499, 41–49, https://doi.org/https://doi.org/10.1016/j.jhydrol.2013.06.008, 2013.
  - Solari, S., Egüen, M., Polo, M. J., and Losada, M. A.: Peaks Over Threshold (POT): A methodology for automatic threshold estimation using goodness of fit p-value, Water Resources Research, 53, 2833–2849, https://doi.org/10.1002/2016WR019426, 2017.
  - Sweet, W. V., Genz, A. S., Obeysekera, J., and Marra, J. J.: A regional frequency analysis of tide gauges to assess Pacific coast flood risk, Frontiers in Marine Science, 7, 1–15, https://doi.org/10.3389/fmars.2020.581769, 2020.